# REGIONVIT: REGIONAL-TO-LOCAL ATTENTION FOR VISION TRANSFORMERS

**Chun-Fu (Richard) Chen, Rameswar Panda, Quanfu Fan**
MIT-IBM Watson AI Lab
`chenrich@us.ibm.com, rpanda@ibm.com, qfan@us.ibm.com`

## ABSTRACT

Vision transformer (ViT) has recently shown its strong capability in achieving comparable results to convolutional neural networks (CNNs) on image classification. However, vanilla ViT simply inherits the same architecture from the natural language processing directly, which is often not optimized for vision applications. Motivated by this, in this paper, we propose a new architecture that adopts the pyramid structure and employ novel regional-to-local attention rather than global self-attention in vision transformers. More specifically, our model first generates regional tokens and local tokens from an image with different patch sizes, where each regional token is associated with a set of local tokens based on the spatial location. The regional-to-local attention includes two steps: first, the regional self-attention extracts global information among all regional tokens and then the local self-attention exchanges the information among one regional token and the associated local tokens via self-attention. Therefore, even though local self-attention confines the scope in a local region but it can still receive global information. Extensive experiments on four vision tasks, including image classification, object and keypoint detection, semantics segmentation and action recognition, show that our approach outperforms or is on par with state-of-the-art ViT variants including many concurrent works. Our source codes and models are available at https://github.com/IBM/RegionViT.

## 1 INTRODUCTION

Transformers (Vaswani et al., 2017) based on self-attention come naturally with the ability to learn long-range dependencies in sequential data. As important as it is to language modeling (Devlin et al., 2019), such ability is also highly desired for many vision tasks where contextual modeling plays a significant role. For this reason, self-attention and transformers have been receiving an increasing attention in the vision community (Bello, 2021; Srinivas et al., 2021; Zhao et al., 2020; Ramachandran et al., 2019a; Bello et al., 2019; Hu et al., 2019; Ramachandran et al., 2019b; Wang et al., 2018). Especially, the recent Vision Transformers (ViT) (Dosovitskiy et al., 2021) demonstrates comparable image classification results against the firmly established and prevalent CNNs in computer vision (He et al., 2016; Tan & Le, 2019; Brock et al., 2021), albeit relying on a huge amount of training data. It has since then led to an explosion of interest in further investigating its potential for a wide variety of vision applications (Wu et al., 2021; Wang et al., 2021; Heo et al., 2021; Zhang et al., 2021a; Li et al., 2021; Graham et al., 2021; Liu et al., 2021; Chu et al., 2021a; Yan et al., 2021; Chen et al., 2021b).

The ViT inherits the entire architecture from the vanilla transformer (Vaswani et al., 2017), which is designed for natural language processing tasks, and some of those designs thus may not meet the needs of vision tasks. For example, the transformer has an isotropic network structure with a fixed number of tokens and unchanged embedding size, which loses the capability to model the context with different scales and allocates computations at different scales. As opposed to this, a majority of CNNs adopt a popular pyramid architecture to compute multi-scale features efficiently. Recent vision transformers such as PVT (Wang et al., 2021) and PiT (Heo et al., 2021) also follow a similar pyramid structure as CNNs, showing improvement on both computation and memory efficiency as well as on model accuracy. Another critical bottleneck of the transformer is that the self-attention module has a quadratic cost in memory and computation with regard to the sequence length (i.e., the number of tokens). This issue is even worse in ViT as images are 2-D, suggesting a quadratic relationship

between the number of tokens and image resolution. As a result, ViT indicates a quadruple complexity w.r.t image resolution. The highly compute- and memory-intensive self-attention makes it challenging to train vision transformer models at fine-grained patch sizes. It also significantly undermines the applications of these models to tasks such as object detection and semantic segmentation, which benefit from or require fine feature information computed from high-resolution images.

To address the aforementioned computational limitations of vision transformers, in this work, we develop a memory-friendly and efficient self-attention method for transformer models to reach their promising potential for vision applications. We propose a novel coarse-to-fine mechanism to compute self-attention in a hierarchical way. Specifically, our approach first divides the input image into a group of non-overlapping patches of large size (e.g., 28×28), on which *regional tokens* are computed via linear projection. Similarly, *local tokens* are created for each region using a smaller patch size (e.g., 4×4). We then use a standard transformer to process regional and local tokens separately. To enable communication between the two types of tokens, we first perform self-attention on regional tokens (*regional attention*) and then jointly attend to the local tokens of each region including their associated regional token (*local attention*). By doing so, regional tokens pass global contextual information to local tokens efficiently while being able to effectively learn from local tokens themselves. For clarity, we represent this two-stage attention mechanism as *Regional-to-Local Attention*, or *R2L attention* for short (see Figure 1 for an illustration). Since both regional and local attention involve much fewer tokens, our R2L attention requires substantially less memory than regular global self-attention used in vision transformers. For example, in our default setting, the memory saving using R2L attention can be up to as much as 73%. We demonstrate the effectiveness of our approach on image classification and several downstream vision tasks including object detection and action recognition.

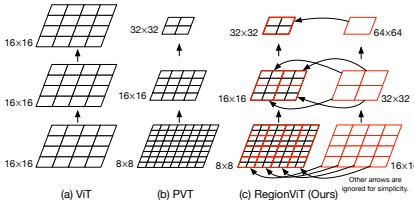

Figure 1: **Regional-to-Local Attention for Vision Transformers.** (a) ViT uses a fixed patch size through the whole network, (b) PVT adopts a pyramid structure to gradually enlarge the patch size in the network. Both ViT and PVT uses all tokens in self-attention, which are computational- and memory-intensive. (c) Our proposed approach combines a pyramid structure with an efficient regional-to-local (R2L) attention mechanism to reduce computation and memory usage. Our approach divides the input image into two groups of tokens, regional tokens of large patch size (red) and local ones of small patch size (black). The two types of tokens communicate efficiently through R2L attention, which jointly attends to the local tokens in the same region and the associated regional token. Note that the numbers denote the patch sizes at each stage of a model.

To summarize, our key contributions in this work are as follows:

1. We propose a new vision transformer (RegionViT) based on regional-to-local attention to learn both local and global features. Our proposed regional-to-local attention alleviates the overhead of standard global attention (too many tokens) and the weakness of pure local attention (no interaction between regions) used in existing vision transformers.

2. Our regional-to-local attention reduces the memory complexity significantly as compared to standard self-attention, leading to a savings in memory complexity by about $\mathcal{O}(N/M^2)$, where $M$ is the window size of a region and $N$ is the total number of tokens. This effectively allows us to train a more deep network for better performance while with comparable complexity.

3. Our models outperform or on par with several concurrent works on vision transformer that exploit pyramid structure for image classification. Experiments also demonstrate that our models work well on several downstream classification tasks, including object detection and action recognition.

## 2    RELATED WORK

**CNNs with Attention.** Attention has been widely used for enhancing the features in CNNs, e.g., SENet (Hu et al., 2018) and CBAM (Woo et al., 2018). Various methods have been proposed that combine self-attention with CNNs (Bello, 2021; Srinivas et al., 2021; Zhao et al., 2020; Ramachandran et al., 2019a; Bello et al., 2019; Hu et al., 2019; Ramachandran et al., 2019b; Wang et al., 2018). E.g., SAN (Zhao et al., 2020) and SASA (Ramachandran et al., 2019a) attempts to replace all convolutional layers with local self-attention network. While these works show promising results, their complexity is relatively high due to the self-attention. On the other hand, BoTNet (Srinivas et al., 2021) replaces

Table 1: **Comparison to related works.** Most works use non-overlapped windows to group tokens, and then propose the corresponding methods to assure the information exchange among regions.

| Methods | Structure | Attention Types | Windows of Local Attention | Handling Non-overlapped Regions | Global Tokens |
|---|---|---|---|---|---|
| ViT (Dosovitskiy et al., 2021) | Isotropic | Global | N/A | N/A | N/A |
| PVT (Wang et al., 2021) | Pyramid | Global | N/A | N/A | N/A |
| Swin (Liu et al., 2021) | Pyramid | Local | Strictly Non-overlapped | Shifting the window | None |
| ViL (Zhang et al., 2021a) | Pyramid | Local + Global | Overlapped | N/A[1] | One learnable token |
| Twins (Chu et al., 2021a) | Pyramid | Local + Global | Non-overlapped | GA | Subsampled from local tokens |
| RegionViT (Ours) | Pyramid | Local + Regional | Non-overlapped | Regional-to-local attention | Regional tokens |

GA: global attention. [1]: not applicable as ViL used overlapped windows.

few convolutional layers with a slightly-modified self-attention to balance computation and accuracy. LambdaNet work (Bello, 2021) uses the approximated self-attention to reduce the overhead of self-attention and make the network efficient. Few works use attention approach to define the regional of interest for the fine-grained visual recognition (Zheng et al., 2020; Ding et al., 2021; Wharton et al., 2021). In contrast, our model utilizes regional-to-local attention to alleviate the workload of global self-attention by local attention while still keeping the global information via regional attention.

**Vision Transformer.** ViT (Dosovitskiy et al., 2021) has recently achieved comparable results to CNNs when using a huge amount of data, e.g., JFT300M (Sun et al., 2017). DeiT (Touvron et al., 2020; Wightman, 2019) subsequently proposes an efficient training scheme that allows vision transformer to work on par with CNNs while training only on ImageNet1K (Deng et al., 2009). Despite promising performance of ViT, the architecture is directly borrowed from natural language processing, which might not be suitable for vision applications. Motivated by this, two main line of research works have been developed to improve ViT. One is to enhance different components of the vision transformer (Yuan et al., 2021; Han et al., 2021; Touvron et al., 2021; Jiang et al., 2021) while still using isotropic structure (i.e., fixed token numbers and channel dimension) like ViT, e.g., while T2T-ViT (Yuan et al., 2021) introduces a Tokens-to-Token (T2T) transformation to encode local structure for each token, CrossViT propose a dual-path architecture, each with different scales, to learn multi-scale features (Chen et al., 2021a). CaiT (Touvron et al., 2021) proposes a layer-scalar to train a deeper network for better performance and LV-ViT (Jiang et al., 2021) modified how the model is trained when CutMix (Yun et al., 2019) augmentation is applied on ViT. Another parallel thread for improving vision transformer is in incorporating CNN-like pyramid structure into ViT (Wu et al., 2021; Wang et al., 2021; Heo et al., 2021; Zhang et al., 2021a; Li et al., 2021; Graham et al., 2021; Liu et al., 2021; Chu et al., 2021a; Yan et al., 2021; Chen et al., 2021b). PVT (Wang et al., 2021) and PiT (Heo et al., 2021) introduce the pyramid structure into ViT, which makes them more suitable for objection detection as it can provide multi-scale features. LocalViT (Li et al., 2021) and ConT (Yan et al., 2021) mix convolutions with self-attention to encode locality information. Swin (Liu et al., 2021), ViL (Zhang et al., 2021a) and Twins (Chu et al., 2021a) limited the self-attention into a local region and then propose different methods to allow the interaction among each local region. Our model also utilizes pyramid structure and limits the self-attention in a local region; while we propose the regional-to-local attention to exchange the information among each region efficiently.

**Difference from PVT, Swin, ViL and Twins.** Table 1 summarizes the difference between our proposed approach, RegionViT with the closely related works. PVT (Wang et al., 2021) uses pyramid structure but the scope of self-attention is still global. While ViL (Zhang et al., 2021a) limits self-attention into a local region, they adopt *overlapping* windows as convolutions to process all the tokens. On the other hand, Swin (Liu et al., 2021), Twins (Chu et al., 2021a) and ours adopt *non-overlapped* windows. While Swin shifts the position of windows alternatively in the consecutive transformer encoder to allow the interaction between regions, Twins subsamples local tokens as the global tokens, and then use the global tokens as the keys for self-attention to achieve the interaction between regions. By contrast, our proposed method utilizes a extra set of regional tokens with regional-to-local attention to perform self-attention on regional tokens only for the global information and then each regional token is sent to the associated local tokens to pass the global information. One major difference from others is that the local self-attention in our approach includes one extra regional token to exchange the global information with local tokens.

## 3 METHOD

Our method is built on top of a vision transformer, so we first present a brief background of ViT and then describe our method (RegionViT) on regional-to-local attention for vision transformers.

Figure 2: **Architecture of the proposed RegionViT**. Two paths are in our proposed network, including regional tokens and local tokens, and their information is exchanged in the regional-to-local (R2L) transformer encoders. In the end, we average all *regional tokens* and use it for the classification. The tensor shape here is computed based on that regional tokens take a patch of $28^2$ and local tokens take a patch of $4^2$.

**Vision Transformer.** As opposed to CNN-based approaches for image classification, ViT is a purely attention-based counterpart, borrowed from NLP. It consists of stacked transformer encoders, each of which contains a multihead self-attention (MSA) and a feed-forward network (FFN) with layer normalization and residual shortcut. To classify an image, ViT first splits it into patches of fixed size, e.g. $16\times16$, and then transforms them into tokens by linear projection. A class token is additionally prepended to the patch tokens to form the input sequence. Before the token are fed into transformer encoders, a learnable absolute positional embedding is added to each token to learn the position information. At the end of the network, the class token is used as the final feature representation for classification. Mathematically, ViT can be expressed as

$$
\begin{aligned}
\mathbf{x}_0 &= [\mathbf{x}_{cls} || \mathbf{x}_{patch}] + \mathbf{x}_{pos} \\
\mathbf{y}_k &= \mathbf{x}_{k-1} + \mathtt{MSA}(\mathtt{LN}(\mathbf{x}_{k-1})), \mathbf{x}_k = \mathbf{y}_k + \mathtt{FFN}(\mathtt{LN}(\mathbf{y}_k)),
\end{aligned}
\tag{1}
$$

where $\mathbf{x}_{cls} \in \mathbb{R}^{1 \times C}$ and $\mathbf{x}_{patch} \in \mathbb{R}^{N \times C}$ are the class token and patch tokens respectively and $\mathbf{x}_{pos} \in \mathbb{R}^{(1+N) \times C}$ is the position embedding, and $[\,||\,]$ denotes the tensor concatenation. $k$, $N$ and $C$ are the layer index, the number of patch tokens and dimension of the embedding, respectively.

While the vanilla ViT is ideally capable of learning global interaction among all the patch tokens, the memory complexity of self-attention becomes high when there are many tokens as the complexity is quadratically linear to the length of the input sequence. Moreover, the use of isotropic structure limits the capability of extending the vanilla ViT model to many vision applications that require high-resolution details, e.g., object detection. To address these issues, we propose regional-to-local attention for vision transformer, RegionViT for short.

**An Overview of Our Proposed Approach (RegionViT).** Figure 2 illustrates the architecture of the proposed vision transformer, which consists of two tokenization processes that convert an image into *regional* (upper path) and *local* tokens (lower path). Each tokenization is a convolution with different patch sizes, e.g., in Figure 2, the patch size of regional tokens is $28^2$ while $4^2$ is used for local tokens with dimensions projected to $C$, which means that one regional token covers $7^2$ local tokens based on the spatial locality, leading to the window size of a local region to $7^2$. At stage 1, two sets of tokens are passed through the proposed regional-to-local transformer encoders. However, for the later stages, to balance the computational load and to have feature maps at different resolutions, we deploy a downsampling process to halve the spatial resolution while doubling the channel dimension like CNN on both regional and local tokens before going to the next stage. Finally, at the end of the network, we average the remaining regional tokens as the final embedding for the classification while the detection uses all local tokens at each stage since it provides more fine-grained location information. By having the pyramid structure, the ViT can generate multi-scale features and hence it could be easily extended to more vision applications, e.g., object detection, rather than image classification only. We explain the main components of the regional-to-local transformer encoder in the next section.

**Regional-to-Local Transformer Encoder.** Figure 3 shows the proposed regional-to-local (R2L) transformer encoder which includes R2L attention and feed-forward network (FFN). Specifically, within R2L attention, the regional self-attention (RSA) first involves all regional tokens to learn the global information efficiently as the number of regional tokens is few and then local self-attention (LSA) takes the local tokens with the associated regional token to learn local feature while the involved regional token could provide global information at the same time. Both RSA and LSA are multihead self-attention (MSA) but with different input tokens. Finally, the FFN is applied to enhance the features. We also add layer normalization (LN) and residual shortcuts as in standard transformer encoders. Mathematically, given the regional and local tokens, $\mathbf{x}_r^{d-1}$ and $\mathbf{x}_l^{d-1}$ as the inputs at layer $d$, the R2L transformer encoder can be expressed as:

$$\mathbf{y}_r^d = \mathbf{x}_r^{d-1} + \text{RSA}(\text{LN}(\mathbf{x}_r^{d-1})), \quad \mathbf{y}_{i,j}^d = [\mathbf{y}_{r_{i,j}}^d || \{\mathbf{x}_{l_{i,j,m,n}}^{d-1}\}_{m,n \in M}]$$

$$\mathbf{z}_{i,j}^d = \mathbf{y}_{i,j}^d + \text{LSA}(\text{LN}(\mathbf{y}_{i,j}^d)), \quad \mathbf{x}_{i,j}^d = \mathbf{z}_{i,j}^d + \text{FFN}(\text{LN}(\mathbf{z}_{i,j}^d)) \quad (2)$$

where $i, j$ are the spatial index with respect to regional tokens while $m, n$ are the index of local token in the window size $M^2$. Note that the input of LSA, $\mathbf{y}_{i,j}^d$, includes one regional token and corresponding local tokens, and thus, the information between local and regional tokens are exchanged; on the other hand, the outputs, $\mathbf{x}_r^d$ and $\mathbf{x}_l^d$, can be extracted from $\mathbf{x}_{i,j}^d$ like the top-right equation.

The RSA exchanges information among all tokens, which covers the context of the whole image; while the LSA combines the features among the tokens belonging to the spatial region, including both regional and local tokens. Since the regions are divided by non-overlapped windows, the RSA is also designed to exchange the information among regions where the LSA takes one regional token and then combines with it the local tokens in the same region. In such a case, all local tokens are still capable of getting global information while being more focused on local neighbors. It is worth noting that the weights are shared be-

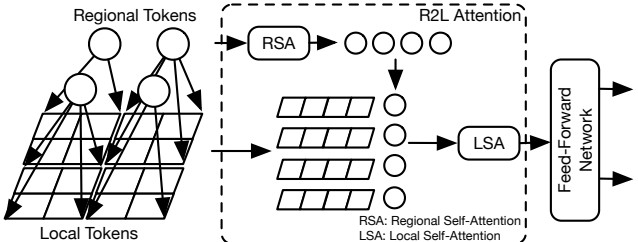

Figure 3: **Illustration of Regional-to-Local (R2L) Transformer Encoder**. All regional tokens are first passed through regional self-attention (RSA) to exchange the information among regions and then local self-attention (LSA) performs parallel self-attention where each takes one regional token and corresponding local tokens. After that, all the tokens are passed through the feed-forward network and split back to the regional and local tokens. RSA and LSA share the same weights.

tween RSA and LSA except for the layer normalization; therefore, the number of parameters won't increase significantly when compared to the standard transformer encoder. With these two attentions, the R2L can effectively and efficiently exchange information among all regional and local tokens. In particular, the self-attention on regional tokens aims to extract high-level information and act as a bridge to pass information of local tokens from one region to other regions. On the other hand, the R2L attention focus on local contextual information with one regional token.

Moreover, the R2L transformer encoder can further reduce the memory complexity significantly. The memory complexity of a self-attention is $\mathcal{O}(N^2)$, where $N$ is the number of local tokens. When a region contains $M$ local tokens, and there are $N/M$ regions, the memory complexity is $\mathcal{O}((M+1)^2 \times (N/M))$; while the complexity from the attention on regional tokens is $\mathcal{O}((N/M)^2)$. Hence, the overall complexity becomes $\mathcal{O}((M+1)^2 \times (N/M) + (N/M)^2)$. E.g., the main component of RegionViT is stage 3, where $M$ is 49 and $N$ is 196; therefore, the memory complexity saving is $\sim 73\%$. The saving of each model and each stage is varied based on the model configuration.

**Relative Position Bias.** The locality is an important clue for understanding visual content; therefore, instead of adding absolute position embedding used by the vanilla ViT, we introduce the relative position bias into the attention map of R2L attention since relative position between patches (or pixels) are more important than the absolution position as objects in the images can be placed in an arbitrary way (Ramachandran et al., 2019a; Liu et al., 2021). We only add this bias to the attention between local tokens and not the attention between regional tokens and local tokens. Specifically, for a given pair of local tokens at location, $(x_m, y_m), (x_n, y_n)$, the attention value $a$ can be expressed as

$$a_{(x_m, y_m),(x_n, y_n)} = \text{softmax}\left( q_{(x_m, y_m)} k_{(x_n, y_n)}^T + b_{(x_m - x_n, y_m - y_n)} \right), \quad (3)$$

where $b_{(x_m - x_n, y_m - y_n)}$ is taken from a learnable parameter $B \in \mathbb{R}^{2M-1 \times 2M-1}$, where $M$ is the window size of a region. The first term $q_{(x_m, y_m)} k_{(x_n, y_n)}^T$ is the attention value based on their content.

With relative position bias, our model does not require the absolute positional embedding as vanilla ViT since this relative position bias helps to encode the position information.

**Downsampling.** The spatial and channel dimension are changed along different stages of our network by simply applying $3 \times 3$ depth-wise convolution with stride 2 for halving the resolution and doubling the channel dimension for both regional and local tokens (weights are shared) (Heo et al., 2021). We also test with regular convolutions but it does not improve the accuracy with increased complexity.

Table 2: Model architectures of RegionViT. For all networks, the dimension per head is 32 and the expanding ratio $r$ in FFN is 4. The patch size of local tokens is always 4 while the patch size of regional tokens is $4 \times M$.

| Model | Tokenization | | Window size ($M$) | Dimension ($C$) | # of Encoders at each stage |
|---|---|---|---|---|---|
| | Local | Regional | | | |
| RegionViT-Ti | 3-conv | linear | 7 | {64, 128, 256, 512} | {2, 2, 8, 2} |
| RegionViT-S | 3-conv | linear | 7 | {96, 192, 384, 768} | {2, 2, 8, 2} |
| RegionViT-M | 1-conv | linear | 7 | {96, 192, 384, 768} | {2, 2, 14, 2} |
| RegionViT-B | 1-conv | linear | 7 | {128, 256, 512, 1024} | {2, 2, 14, 2} |

1-conv: Conv(out=C, k=8, s=4, p=3). 3-conv: Conv(out=C/4, k=3, s=2, p=1) → Conv(out=C/2, k=3, s=2, p=1) →
Conv(out=C, k=3, s=1, p=1), where C is the channel dimension of the first block. LayerNorm and GeLU are added between Conv.

**Input Tokenization.** The tokenization for local tokens can be implemented by using a $4 \times 4$ convolution with channel size $C$ and stride 4 (i.e. non-overlapped). However, as pointed out in T2T (Yuan et al., 2021), CrossViT (Chen et al., 2021a), and PiT (Heo et al., 2021), using a stronger but still simple subnetwork for the tokenization could further improve the performance, especially for the smaller models. Thus, we adopt two input tokenizations in our model, one still contains only one convolutional layer and another one contains three convolutional layers (see Table 2 for details).

**Regional Tokens.** We adopt a simple approach to generate regional tokens, that is, using linear projection with larger patch sizes in comparison to the patch size for the local tokens. E.g., as shown in Figure 2, the patch size for local tokens is $4 \times 4$, we simply use patch size $28 \times 28$ to split the image and then project each patch linearly to generate regional tokens.

Table 2 shows our RegionViT models with different configurations. By default, the window size is 7 while we also experiment with a larger window size (14), those models are annotated with +.

## 4 EXPERIMENTS

### 4.1 IMAGE CLASSIFICATION

**Datasets.** We use ImageNet1K (Deng et al., 2009) (IN1K) and ImageNet21K (Deng et al., 2009) (IN21K) to validate our method. ImageNet1K contains 1.28 million training images and 50k validation images over 1k classes, and ImageNet21K is a large-scale dataset that consists of around 14 million images over 21,841 classes. We use all images for training and then finetune the model on ImageNet1K. Moreover, we also perform the transfer learning from ImageNet1K to five downstream datasets, including CIFAR10 (Krizhevsky et al., 2009), CIFAR100 (Krizhevsky et al., 2009), IIIPets (Parkhi et al., 2012), StandfordCars (Krause et al., 2013) and ChestXRay8 (Wang et al., 2017).

**Training and Evaluation.** We follow DeiT (Touvron et al., 2020) to train our models on IN1K except that we use batch size 4,096 with a base learning rate 0.004 and the warm-up epochs is 50. We adopt the AdamW (Loshchilov & Hutter, 2019) optimizer with cosine learning rate scheduler (Loshchilov & Hutter, 2017). We apply Mixup (Zhang et al., 2018), CutMix (Yun et al., 2019), RandomErasing (Zhong et al., 2020), label smoothing (Szegedy et al., 2016), RandAugment (Cubuk et al., 2020) and instance repetition (Hoffer et al., 2020). During training, we random cropped a $224 \times 224$ region and take a $224 \times 224$ center crop after resizing the shorter side to 256 for evaluation. We used a similar setting for IN21K and transfer learning, and more details can be found in Section A.1.

Table 3: Comparisons with recent pyramid-like structure-based ViT models on ImageNet1K. The bold numbers indicate the best number within each section.

| Model | Params (M) | FLOPs (G) | Acc. (%) | Model | Params (M) | FLOPs (G) | Acc. (%) |
|---|---|---|---|---|---|---|---|
| PiT-XS (Heo et al., 2021) | 10.6 | 1.4 | 78.1 | ConT-M (Yan et al., 2021) | 39.6 | 6.4 | 81.8 |
| ConT-S (Yan et al., 2021) | 10.1 | 1.5 | 76.5 | Twins-PCPVT-B (Chu et al., 2021a) | 43.8 | 6.4 | 82.7 |
| PVT-T (Wang et al., 2021) | 13.2 | 1.9 | 75.1 | PVT-M (Wang et al., 2021) | 44.2 | 6.7 | 81.2 |
| ConViT-Ti+ (d'Ascoli et al., 2021) | 10.0 | 2.0 | 76.7 | CvT-21 (Wu et al., 2021) | 32.0 | 7.1 | 82.5 |
| Twins-SVT-S (Chu et al., 2021a) | 24.0 | 2.8 | **81.7** | Twins-SVT-B (Chu et al., 2021a) | 56 | 8.3 | 83.2 |
| PiT-S (Heo et al., 2021) | 23.5 | 2.9 | 80.9 | ViL-M (Zhang et al., 2021a) | 39.7 | 8.7 | 83.3 |
| ConT-M (Yan et al., 2021) | 19.2 | 3.1 | 80.2 | Swin-S (Liu et al., 2021) | 50.0 | 8.7 | 83.0 |
| Twins-PCPVT-S (Chu et al., 2021a) | 24.1 | 3.7 | 81.2 | PVT-L (Wang et al., 2021) | 61.4 | 9.8 | 81.7 |
| PVT-S (Wang et al., 2021) | 24.5 | 3.8 | 79.8 | ConViT-S+ (d'Ascoli et al., 2021) | 48.0 | 10.0 | 82.2 |
| | | | | NesT-S (Zhang et al., 2021b) | 38.0 | 10.4 | 83.3 |
| RegionViT-Ti | 13.8 | 2.4 | 80.4 | RegionViT-M | 41.2 | 7.4 | 83.1 |
| RegionViT-Ti+ | 14.3 | 2.7 | 81.5 | RegionViT-M+ | 42.0 | 7.9 | **83.4** |
| DeiT-S (Touvron et al., 2020) | 22.1 | 4.6 | 79.9 | DeiT-B (Touvron et al., 2020) | 86.6 | 17.6 | 81.8 |
| CvT-13 (Wu et al., 2021) | 20.0 | 4.5 | 81.6 | PiT-B (Heo et al., 2021) | 73.8 | 12.5 | 82.0 |
| Swin-T (Liu et al., 2021) | 29.0 | 4.5 | 81.3 | ViL-B (Zhang et al., 2021a) | 55.7 | 13.4 | 83.2 |
| LocalViT-S (Li et al., 2021) | 22.4 | 4.6 | 80.8 | Twins-SVT-L (Chu et al., 2021a) | 99.2 | 14.8 | 83.7 |
| ViL-S (Zhang et al., 2021a) | 24.6 | 4.9 | 82.0 | Swin-B (Liu et al., 2021) | 88.0 | 15.4 | 83.5 |
| Visformer-S (Chen et al., 2021b) | 40.2 | 4.9 | 82.3 | ConViT-B (d'Ascoli et al., 2021) | 86.0 | 17.0 | 82.4 |
| ConViT-S (d'Ascoli et al., 2021) | 27.0 | 5.4 | 81.3 | NesT-B (Zhang et al., 2021b) | 68.0 | 17.9 | **83.8** |
| NesT-S (Zhang et al., 2021b) | 17.0 | 5.8 | 81.5 | | | | |
| RegionViT-S | 30.6 | 5.3 | 82.6 | RegionViT-B | 72.7 | 13.0 | 83.2 |
| RegionViT-S+ | 31.3 | 5.7 | **83.3** | RegionViT-B+ | 73.8 | 13.6 | **83.8** |

Table 4: Results on IN1K with IN21K and transfer learning.

(a) IN1K results with IN21K.

| Model | Params (M) | FLOPs (G) | Acc. (%) |
|---|---|---|---|
| ViL-B | 55.7 | 43.7 | 86.0 |
| Swin-L | 197.0 | 103.9 | 87.3 |
| CvT-W24 | 277 | 193.2 | **87.7** |
| RegionViT-B+ | 76.5 | 42.6 | 86.5 |

(b) Transfer learning on downstream tasks.

| | CIFAR10 | CIFAR100 | Pet | StanfordCars | ChestXRay8 |
|---|---|---|---|---|---|
| DeiT-S | 99.1 | 90.9 | 94.9 | 91.5 | 55.4 |
| DeiT-B | 99.1 | 90.8 | 94.4 | 91.7 | 55.8 |
| RegionViT-S | 98.9 | 90.0 | 95.3 | 92.8 | 57.8 |
| RegionViT-M | 99.0 | 90.8 | 95.5 | 91.9 | 58.3 |

Table 5: Object detection performance on MS COCO val2017 with 1× and 3× schedule. The bold number indicates the best number within the section, and for MaskRCNN, both $AP^b$ and $AP^m$ are annotated.

| | Params | FLOPs | RetineNet | | Params | FLOPs | MaskRCNN | | | |
|---|---|---|---|---|---|---|---|---|---|---|
| | | | 1× | 3× | | | 1× | | 3× | |
| Backbone | (M) | (G) | $AP$ | $AP$ | (M) | (G) | $AP^b$ | $AP^m$ | $AP^b$ | $AP^m$ |
| ResNet50 | 37.7 | 234 | 36.3 | 39.0 | 44.2 | 260.0 | 38.0 | 34.4 | 41.0 | 37.1 |
| ConT-M (Yan et al., 2021) | 27.0 | 217.2 | 39.3 | – | – | – | 40.5 | 38.1 | – | – |
| PVT-S (Wang et al., 2021) | 34.2 | – | 40.4 | 42.2 | 44.1 | – | 40.4 | 37.8 | 43.0 | 39.6 |
| ViL-S (Zhang et al., 2021a) | 35.7 | 252.2 | 41.6 | 42.9 | 45.0 | 174.3 | 41.8 | 38.5 | 43.4 | 39.6 |
| Swin-T (Liu et al., 2021) | 38.5 | 245 | 41.5 | 43.9 | 47.8 | 264 | 42.2 | 39.1 | 46.0 | 41.6 |
| Twins-SVT-S (w/ PEG) (Chu et al., 2021a) | 34.3 | 209 | 43.0 | 45.6 | 44.0 | 228 | 43.5 | 40.3 | 46.8 | 42.6 |
| RegionViT-S | 40.8 | 192.6 | 42.2 | 45.8 | 50.1 | 171.3 | 42.5 | 39.5 | 46.3 | 42.3 |
| RegionViT-S+ | 41.5 | 204.2 | 43.1 | **46.9** | 50.9 | 182.9 | 43.5 | 40.4 | 47.3 | **43.4** |
| RegionViT-S+ w/ PEG | 41.6 | 204.3 | **43.9** | 46.7 | 50.9 | 183.0 | **44.2** | **40.8** | **47.6** | **43.4** |
| ResNet101 | 56.7 | 315 | 38.5 | 40.9 | 63.2 | 336 | 40.4 | 36.4 | 42.8 | 38.5 |
| ResNeXt101-32x4d | 56.4 | 319.1 | 39.9 | 41.4 | 62.8 | 340 | 41.9 | 37.5 | 44.0 | 39.2 |
| PVT-M (Wang et al., 2021) | 53.9 | – | 41.9 | 43.2 | 63.9 | – | 42.0 | 39.0 | 44.2 | 40.5 |
| ViL-M (Zhang et al., 2021a) | 50.8 | 338.9 | 42.9 | 43.7 | 60.1 | 261.1 | 43.4 | 39.7 | 44.6 | 40.7 |
| Swin-S (Liu et al., 2021) | 59.8 | 335 | 44.5 | 46.3 | 69.1 | 354 | 44.8 | 40.9 | 47.6 | 42.8 |
| Twins-SVT-B (w/ PEG) (Chu et al., 2021a) | 67.0 | 322 | **45.3** | 46.9 | 76.3 | 340 | 45.2 | 41.5 | 48.0 | 43.0 |
| RegionViT-B | 83.4 | 308.9 | 43.3 | 46.1 | 92.2 | 287.9 | 43.5 | 40.1 | 47.2 | 43.0 |
| RegionViT-B+ | 84.4 | 328.1 | 44.2 | **46.9** | 93.2 | 307.1 | 44.5 | 41.0 | 48.1 | 43.5 |
| RegionViT-B+ w/ PEG | 84.5 | 328.2 | 44.6 | **46.9** | 93.2 | 307.2 | **45.4** | **41.6** | **48.3** | **43.5** |
| ResNeXt101-64x4d | 95.5 | 473 | 41.0 | – | 101.9 | 493 | 42.8 | 38.4 | – | – |
| PVT-L (Wang et al., 2021) | 71.1 | 345 | 42.6 | – | 81.0 | 364 | 42.9 | 39.5 | – | – |
| ViL-B (Zhang et al., 2021a) | 66.7 | 443.0 | 44.3 | 44.7 | 76.1 | 365.1 | 45.1 | 41.0 | 45.7 | 41.3 |
| Swin-B (Liu et al., 2021) | 98.4 | 477 | 44.7 | – | 107.2 | 496 | 45.5 | 41.3 | – | – |
| Twins-SVT-L (w/ PEG) (Chu et al., 2021a) | 110.9 | 455 | 45.7 | – | 119.7 | 474 | 45.9 | 41.6 | – | – |
| RegionViT-B+ w/ PEG† | 84.5 | 506.4 | **46.1** | **48.2** | 93.2 | 464.4 | **46.3** | **42.4** | **49.2** | **44.5** |

The reported results of Swin are from Twins as the original paper does not include resutls with ImageNet1K weights. †: input resolution is 896×1344.

**Results on ImageNet1K.** Table 3 shows the results on ImageNet1K where the methods listed all adopt CNN-like pyramid structure into the ViT and are all very recent and concurrent works. For the smaller models (Ti, S), RegionViT achieves a better trade-off between accuracy and complexity (parameters or FLOPs); on the other hand, for the larger models (M and B), it obtains better accuracy while having fewer FLOPs and parameters. The efficiency of RegionViT comes from the proposed R2L transformer encoder, and hence with such efficiency, it enables the network to be wide and deep for better accuracy with comparable complexity.

**Results on ImageNet21K.** Table 4a shows that our model achieves a good accuracy-efficiency tradeoff with ImageNet21K for pretraining. RegionViT-B+ only needs 25% parameters and FLOPs compared to CvT-W24 (Wu et al., 2021) and half parameters and FLOPs to Swin-L (Liu et al., 2021).

**Results on Transfer Learning.** Table 4b shows the transfer learning results with ImageNet1K pretrained weights. Our model outperforms DeiT (Touvron et al., 2020) by 2∼3% on ChestXRay8, which has a larger domain gap from ImageNet1K than other four datasets. We think this is because the hierarchical feature models could provide better generalization for the domain gap compared to DeiT which uses isotropic spatial resolution throughout the whole network.

## 4.2 OBJECT DETECTION AND SEMANTIC SEGMENTATION

**Dataset.** We use MS COCO 2017 to validate our models (Lin et al., 2014) on object detection. COCO 2017 dataset contains 118K images for training and 5K images for validation across 80 categories. We also test our model on keypoint detection and results are shown in Sec. B. On the other hand, we validate our model with semantic segmentation on ADE20K (Zhou et al., 2017). The ADE20K dataset contains 20k images in the training set, 2k images for validation.

**Training and Evaluation.** For object detection, we adopt RetinaNet (Lin et al., 2017) and MaskR-CNN (He et al., 2017) as our detection framework. We simply replace the backbone with RegionViT, and then output the local tokens at each stages as multi-scale features for the detector. The regional

tokens are not used in the detection. Before sending the features to the detector framework, we add new layer normalization layers to normalize the features. We use RegionViT-S and RegionViT-B as the backbone and the weights are initialized from ImageNet1K pretraining. The shorter side and longer side of the input image is resized to 672 and 1,120, respectively. We train our models based on the settings in $1\times$ and $3\times$ schedule in Detectron2 (Wu et al., 2019) for object detection. More training details could be found in A.2. For semantic segmentation, we adopt Semantic FPN as the framework (Kirillov et al., 2019) and use RegionViT as the backbone. We initialize models with ImageNet1K weights, and mostly follow the training receipt in Twins' paper (Chu et al., 2021a). More training details can be found in Sec. A.3.

**Results on Object Detection.** Table 5 compares RegionViT with other methods. For the smaller models, RegionViT-S and RegionViT-S+ achieve similar accuracy while being moderately efficient in terms of FLOPs. We find that the position encoding generator (PEG) proposed by (Chu et al., 2021b) could help the detection accuracy substantially, so we also provide the results with PEG, which are also used in Twins (Chu et al., 2021a). For the middle size models with RetinaNet, RegionViT-B are slightly worse than Twins with $1\times$ schedule; however, our model catches up the performance with $3\times$ schedule. When comparing the large models under similar FLOPs (RegionViT-B+ w/ PEG†), we further train our model with the similar resolution used by other works, and observe that our models outperform all others while being more parameter efficient (models marked with † in Table 5).

**Results on Semantic Segmentation.** Table 6 compares ours to Swin (Liu et al., 2021) and Twins. As PEG is useful for object detection, we show the results with PEG. Our models achieved significant improvement with similar FLOPs for both models. This suggests that the proposed R2L attention can effectively model global context.

Table 6: Performance on semantic segmentation.

| Model | FLOPs (G) | Params (M) | mIoU (%) |
|---|---|---|---|
| Swin-T* | 46 | 31.9 | 41.5 |
| Twins-SVT-S | 37 | 28.3 | 43.2 |
| RegionViT-S+ (w/ PEG) | 37 | 35.7 | 45.3 |
| Swin-S* | 70 | 53.2 | 45.2 |
| Twins-SVT-B | 67 | 60.4 | 45.3 |
| RegionViT-B+ (w/ PEG) | 67 | 78.3 | 47.5 |

*: Numbers are cited from the reproduced results of Twins' paper.

## 4.3 ACTION RECOGNITION

**Datasets and Setup.** We validate our approach on Kinetics400 (K400) (Kay et al., 2017) and Something-Something V2 (SSV2) (Goyal et al., 2017). We adopt the divided-space-time attention in TimeSformer (Bertasius et al., 2021) as the temporal modeling to perform action recognition experiments. More details could be found in Sec. A.4.

**Results.** Table 7 shows that with RegionViT as the backbone, the model can be much more efficient with competitive accuracy to TimeSformer, which uses vanilla ViT as the backbone. RegionViT-M could reduce more than 50% FLOPs and parameters than the TimeS-

Table 7: Performance on action recognition.

| Model | FLOPs* (G) | Params (M) | K400 Acc. (%) | SSV2 Acc. (%) |
|---|---|---|---|---|
| TimeSformer | 197 | 121.4 | 75.8 | 59.5 |
| x TimeSformer† | 197 | 121.4 | 77.1 | 59.2 |
| RegionViT-S | 59.4 | 42.9 | 76.6 | 59.7 |
| RegionViT-M | 83.1 | 57.5 | **77.6** | **59.8** |

*: FLOPs of single crop, †: retrained with the same setting.

former. This not only shows the importance of spatial modeling in action recognition but also validates that our proposed model can also be extended for efficient action recognition.

## 4.4 ABLATION STUDY

We perform the following experiments to verify the effectiveness of different components in Region-ViT. All experiments are conducted based on RegionViT-S.

**Regional Tokens.** Table 8 shows the results with and without regional tokens on three tasks, and Table A5 includes FLOPs and parameters comparison. For image classification, the regional tokens provide around 0.4% improvement with negligible overhead in both computations (6%) and parameters (0.7%). On the other hand, the regional tokens clearly improve the

Table 8: Ablation on regional tokens.

| Dataset | IN1K | MS COCO | | ADE20K |
|---|---|---|---|---|
| Regional Tokens | Acc. | MaskRCNN ($AP^b/AP^m$) | RetinaNet ($AP^b$) | SemanticFPN (mIoU) |
| N | 82.2 | 40.5/37.8 | 40.7 | 42.3 |
| Y | 82.6 | 42.5/39.5 | 42.2 | 43.7 |

performance of object detection and semantic segmentation with negligible overhead ($<2\%$ on both FLOPs and parameters). This is because dense prediction tasks require more multi-scale features with global contextual information (provided by the regional tokens) than image classification.

**Downsampling.** Table 9 shows different approaches for downsampling the patches. 3×3 kernel size achieves better results than 2×2 convolution, because the kernel is non-overlapped with 2×2 convolution, which limits the interaction among local tokens. Moreover, using regular convolution does not improve the performance but increases computations and parameters. Thus, we use 3×3 depthwise convolution for downsampling.

Table 9: Different downsampling approaches.

| Downsampling | Params | FLOPs (G) | IN1K Acc. (%) |
|---|---|---|---|
| 2×2 convolution | 32.1 | 5.5 | 82.3 |
| 3×3 convolution | 34.1 | 5.7 | 82.5 |
| 3×3 depth-wise conv. | 30.6 | 5.3 | 82.6 |

**Weight sharing between RSA and LSA as well as downsampling.** Table 10 shows the results with and without weight sharing. As seen from the table, using separated weights for RSA and LSA only slightly improves the accuracy but significantly increases the model size. This suggests sharing parameters between RSA and LSA suffices for learning local and global information in our approach.

Table 10: Weight sharing.

| Weight sharing | Params | FLOPs (G) | IN1K Acc. (%) |
|---|---|---|---|
| N | 40.4 | 5.3 | 82.7 |
| Y | 30.6 | 5.3 | 82.6 |

**Keys in LSA.** Table 11 studies the number of regional tokens in LSA. When using all regional tokens in the LSA, it provides the local tokens the possibility to explore all global information. Nonetheless, this model only improves 0.1% but with 13% more computations, which suggests that when considering the regional tokens, the one associated with the current region is sufficient to achieve good performance.

Table 11: Keys in LSA.

| Regional tokens | Params | FLOPs (G) | IN1K Acc. (%) |
|---|---|---|---|
| All | 30.6 | 6.0 | 82.7 |
| Corresponding | 30.6 | 5.3 | 82.6 |

**Position information.** Table 12 compares performance of different combinations of absolute position embedding (APE) and relative position bias (RPB). The models with RPB achieve better accuracy with similar FLOPs and parameters. Although the model with APE could slightly improve the performance, it limits the model to run at a fixed resolution, which is not suitable for vision tasks where image size could be varied. Thus, we only adopt the relative position bias in our models.

Table 12: Different position information.

| Abs. pos. | Rel. pos. | Params (M) | FLOPs (G) | IN1K Acc. (%) |
|---|---|---|---|---|
| N | N | 30.6 | 5.3 | 82.4 |
| Y | N | 30.9 | 5.3 | 82.2 |
| Y | Y | 30.9 | 5.3 | 82.7 |
| N | Y | 30.6 | 5.3 | 82.6 |

**Overlapped windows.** Table 13 compares the results with and without overlapped windows. The overlapping ratio of neighboring windows is 50% and hence this model allows more interactions among local tokens. It improves RegionViT-S by 0.2% but increases the FLOPs by 3.5×. This suggests that the information exchange between the border of windows is sufficiently covered by the proposed R2L attention.

Table 13: Overlapped windows.

| Model | Params | FLOPs (G) | IN1K Acc. (%) |
|---|---|---|---|
| overlapped | 30.6 | 18.8 | 82.8 |
| non-overlapped | 30.6 | 5.3 | 82.6 |

**Comparison between R2L attention and shifted-window attention.** Table 14 shows the comparison between shifted-window attention (Liu et al., 2021) and proposed R2L attention. R2L attention outperforms shifted-window attention by 0.7%, which shows that our proposed R2L attention can effectively model global information to achieve good performance.

Table 14: Comparison of different attentions.

| Model | Params | FLOPs (G) | IN1K Acc. (%) |
|---|---|---|---|
| Shifted-window | 29.0 | 4.5 | 81.3 |
| R2L | 32.1 | 5.2 | 82.0 |

## 5  CONCLUSION

In this paper, we propose a new ViT architecture that exploits the pyramid structure used in most of CNNs to provide multi-scale features and hence, the models can be more easily extended to different vision applications, like object detection. Moreover, the proposed regional-to-local attention relaxes the memory overhead of performing self-attention on the fine-grained image tokens by limiting the scope of attention but still keeping the capability to explore global information. Extensive experiments on several standard benchmark datasets well demonstrate that our proposed models outperform or are on par with many concurrent ViT variants on four vision applications, including image classification, object and keypoint detection, semantic segmentation and action recognition.

## ETHICS STATEMENT

Our work introduces a memory-friendly and efficient self-attention method for transformer models, which have wide-ranging applications to image classification, object detection, human activity recognition, etc. By improving efficiency, our work can have a positive effect on these applications in society, allowing faster processing. E.g., this could enable faster responses to detected events such as medical emergencies or detecting defects in manufacturing parts. Lower computation costs could also save energy and therefore have a positive impact on the environment. Negative impacts of our research are difficult to predict, however, it shares many of the pitfalls associated with deep classification models. E.g., Image and video classification systems have negative implications on privacy and could be used by malicious actors or governments to infringe on the privacy of citizens. Future research into private and ethical aspects of visual recognition is an important direction.

## REPRODUCIBILITY STATEMENT

In order to reproduce our results, we describe the details of the hyperparameters used in our training in Appendix A for all the tasks and we also attach our codes for image classification as part of the supplemental material. We will publicly release all the codes and models after acceptance. In the attached codes, we set all the hyperparameters as their default value, so that users can re-run our experiments with exactly the same hyperparameters. The README file also describes the requirement to build necessary Python environment for running the codes.

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

Table A1: **Details of training settings for image classification on ImageNet1K and ImageNet21K.**

| | IN1K | IN21K | Finetune to IN1K@$384^2$ | Transfer |
|---|---|---|---|---|
| Batch size | 4,096 | 4,096 | 1,024 | 768 |
| Epochs | 300 | 120 | 30 | 1,000 |
| Optimizer | AdamW | AdamW | AdamW | SGD |
| Weight Decay | 0.05 | 0.01 | 1e-8 | 1e-4 |
| Linear-rate Scheduler (Initial LR) | Cosine (0.004) | Cosine (0.001) | Cosine (0.0002) | Cosine (0.01) |
| Warmup Epochs | 50 | 5 | 0 | 5 |
| Warmup linear-rate Scheduler (Initial LR) | | | Linear (1e-6) | |
| Data Aug. | | | RandAugment (m=9, n=2) | |
| Mixup ($\alpha$) | | | 0.8 | |
| CutMix ($\alpha$) | | | 1.0 | |
| Random Erasing | | | 0.25 | 0.0 |
| Instance Repetition* | | | 3 | |
| Drop-path | | | 0.1 | 0.0 |
| Label Smoothing | | | 0.1 | |

*: disabled for RegionViT-Ti.

## APPENDIX

**Summary** This appendix contains the following additional details. First, in Sec. A, we describe the training details on image classification, object and keypoint detection, semantic segmentation and action recognition separately. Then, we provide detailed results on object detection and ablation studies. The supplementary materials include our codes to reproduce our results on image classification and the README file in the attached codes (**RegionViT_Code.zip**) also provides the detailed instructions to train and evaluate the networks.

## A TRAINING DETAILS

### A.1 IMAGE CLASSIFICATION

We follow DeiT (Touvron et al., 2020) to train our models on ImageNet1K (Deng et al., 2009) except that we use batch size 4,096 with base learning rate 0.004 and the warm-up epochs is 50. We adopt the AdamW (Loshchilov & Hutter, 2019) optimizer with cosine learning rate scheduler (Loshchilov & Hutter, 2017). We apply Mixup (Zhang et al., 2018), CutMix (Yun et al., 2019), RandomErasing (Zhong et al., 2020), label smoothing (Szegedy et al., 2016), RandAugment (Cubuk et al., 2020) and instance repetition (Hoffer et al., 2020). During training, we randomly crop a 224×224 region and take a 224×224 center crop after resizing the shorter side to 256 for evaluation. While pretraining on ImageNet21K (Deng et al., 2009), we use similar settings with slight modifications. We train the model using 120 epochs with 5 epochs for warmup, weight-decay of 0.01 and base learning rate of 0.001. For transfer learning experiments, we finetune the ImageNet1K pretrained models with 1,000 epochs, batch size of 768, learning rate of 0.01, SGD optimizer, weight decay of 0.0001, and using the same data augmentation in training on ImageNet1K. During evaluation, we resize the shorter side of an image to 256 and then take a 224×224 region at the center and report top-1 accuracy. For finetuning experiments from ImageNet21K, we finetune the models with higher resolution of 384×384, for 30 epochs with cosine learning rate scheduler, base linear rate of 0.002, weight-decay of 1e-8. Moreover, as it is trained on larger resolution, we adjust the window size $M$ to have the same number of regional tokens at the first stage, e.g., we change the window size to 12 for the the models trained at 224×224 with window size 7. We adopt the bicubic interpolation to upsample the weights of tokenizations and relative position bias. During evaluation, we directly resize the shorter side to 384 and then take center 384×384 crop. We trained the models with 32 GPUs.

Table A2: Object detection performance on the COCO val2017 with 1× schedule. The bold number indicates the best number within the section, and for MaskRCNN, both $AP^b$ and $AP^m$ are annotated.

| Backbone | Params (M) | FLOPs (G) | RetineNet AP | $AP_{50}$ | $AP_{75}$ | $AP_S$ | $AP_M$ | $AP_L$ | Params (M) | FLOPs (G) | MaskRCNN $AP^b$ | $AP^b_{50}$ | $AP^b_{75}$ | $AP^m$ | $AP^m_{50}$ | $AP^m_{75}$ |
|---|---|---|---|---|---|---|---|---|---|---|---|---|---|---|---|---|
| ResNet50 | 37.7 | 234 | 36.3 | 55.3 | 38.6 | 19.3 | 40.0 | 48.8 | 44.2 | 260.0 | 38.0 | 58.6 | 41.4 | 34.4 | 55.1 | 36.7 |
| ConT-M (Yan et al., 2021) | 27.0 | 217.2 | 39.3 | 59.3 | 41.8 | 23.1 | 43.1 | 51.9 | – | – | 40.5 | – | – | 38.1 | | |
| PVT-S (Wang et al., 2021) | 34.2 | – | 40.4 | 61.3 | 43.0 | 25.0 | 42.9 | 55.7 | 44.1 | – | 40.4 | 62.9 | 43.8 | 37.8 | 60.1 | 40.3 |
| ViL-S (Zhang et al., 2021a) | 35.7 | 252.2 | 41.6 | 62.5 | 44.1 | 24.9 | 44.6 | 56.2 | 45.0 | 174.3 | 41.8 | 64.1 | 45.1 | 38.5 | 61.1 | 41.4 |
| Swin-T (Liu et al., 2021) | 38.5 | 245 | 41.5 | 62.1 | 44.2 | 25.1 | 44.9 | 55.5 | 47.8 | 264 | 42.2 | 64.6 | 46.2 | 39.1 | 61.6 | 42.0 |
| Twins-SVT-S (w/ PEG) (Chu et al., 2021a) | 34.3 | 209 | 43.0 | 64.2 | 46.3 | 28.0 | 46.4 | 57.5 | 44.0 | 228 | 43.5 | 66.0 | 47.3 | 40.3 | 63.2 | 43.4 |
| RegionViT-S | 40.8 | 192.6 | 42.2 | 64.1 | 45.1 | 27.5 | 45.4 | 55.3 | 50.1 | 171.3 | 42.5 | 65.8 | 46.1 | 39.5 | 62.8 | 42.2 |
| RegionViT-S+ | 41.5 | 204.2 | 43.1 | 64.8 | 46.2 | 29.6 | 46.6 | 56.1 | 50.9 | 182.9 | 43.5 | 66.9 | 47.5 | 40.4 | 63.7 | 43.4 |
| RegionViT-S+ w/ PEG | 41.6 | 204.3 | **43.9** | 65.5 | 47.3 | 28.5 | 47.3 | 57.9 | 50.9 | 183.0 | **44.2** | 67.3 | 48.2 | **40.8** | 64.1 | 44.0 |
| ResNet101 | 56.7 | 315 | 38.5 | 57.8 | 41.2 | 21.4 | 42.6 | 51.1 | 63.2 | 336 | 40.4 | 61.1 | 44.2 | 36.4 | 57.7 | 38.8 |
| ResNeXt101-32x4d | 56.4 | 319.1 | 39.9 | 59.6 | 42.7 | 22.3 | 44.2 | 52.5 | 62.8 | 340 | 41.9 | 62.5 | 45.9 | 37.5 | 59.4 | 40.2 |
| PVT-M (Wang et al., 2021) | 53.9 | – | 41.9 | 63.1 | 44.3 | 25.0 | 44.9 | 57.6 | 63.9 | – | 42.0 | 64.4 | 45.6 | 39.0 | 61.6 | 42.1 |
| ViL-M (Zhang et al., 2021a) | 50.8 | 338.9 | 42.9 | 64.0 | 45.4 | 27.0 | 46.1 | 57.2 | 60.1 | 261.1 | 43.4 | 65.9 | 47.0 | 39.7 | 62.8 | 42.1 |
| Swin-S (Liu et al., 2021) | 59.8 | 335 | 44.5 | 65.7 | 47.5 | 27.4 | 48.0 | 59.9 | 69.1 | 354 | 44.8 | 66.6 | 48.9 | 40.9 | 63.4 | 44.2 |
| Twins-SVT-B (w/ PEG) (Chu et al., 2021a) | 67.0 | 322 | **45.3** | 66.7 | 48.1 | 28.5 | 48.9 | 60.6 | 76.3 | 340 | 45.2 | 67.6 | 49.3 | 41.5 | 64.5 | 44.8 |
| RegionViT-B | 83.4 | 308.9 | 43.3 | 65.2 | 46.4 | 29.2 | 46.4 | 57.0 | 92.2 | 287.9 | 43.5 | 66.7 | 47.4 | 40.1 | 63.4 | 43.0 |
| RegionViT-B+ | 84.4 | 328.1 | 44.2 | 66.2 | 47.1 | 29.2 | 47.5 | 58.6 | 93.2 | 307.1 | 44.5 | 67.6 | 48.7 | 41.0 | 64.4 | 43.9 |
| RegionViT-B+ w/ PEG | 84.5 | 328.2 | 44.6 | 66.4 | 47.6 | 29.6 | 47.6 | 59.0 | 93.2 | 307.2 | **45.4** | 68.4 | 49.6 | **41.6** | 65.2 | 44.8 |
| ResNeXt101-64x4d | 95.5 | 473 | 41.0 | 60.9 | 44.0 | 23.9 | 45.2 | 54.0 | 101.9 | 493 | 42.8 | 63.8 | 47.3 | 38.4 | 60.6 | 41.3 |
| PVT-L (Wang et al., 2021) | 71.1 | 345 | 42.6 | 63.7 | 45.4 | 25.8 | 46.0 | 58.4 | 81.0 | 364 | 42.9 | 65.0 | 46.6 | 39.5 | 61.9 | 42.5 |
| ViL-B (Zhang et al., 2021a) | 66.7 | 443.0 | 44.3 | 65.5 | 47.1 | 28.9 | 47.9 | 58.3 | 76.1 | 365.1 | 45.1 | 67.2 | 49.3 | 41.0 | 64.3 | 44.2 |
| Swin-B (Liu et al., 2021) | 98.4 | 477 | 44.7 | 65.9 | 49.2 | – | – | – | 107.2 | 496 | 45.5 | – | – | 41.3 | – | – |
| Twins-SVT-L (w/ PEG) (Chu et al., 2021a) | 110.9 | 455 | 45.7 | 67.1 | 49.2 | – | – | – | 119.7 | 474 | 45.9 | – | – | 41.6 | – | – |
| RegionViT-B+ w/ PEG† | 84.5 | 506.4 | **46.1** | 68.0 | 49.5 | 30.5 | 49.9 | 60.1 | 93.2 | 464.4 | **46.3** | 69.1 | 51.2 | **42.4** | 66.2 | 45.6 |

The reported results of Swin (Liu et al., 2021) are from Twins (Chu et al., 2021a) as the original paper does not include resutls with ImageNet1K weights. †: input resolution is 896×1344.

## A.2 OBJECT DETECTION AND KEYPOINT DETECTION

We adopt RetinaNet (Lin et al., 2017) and MaskRCNN (He et al., 2017) as our detection framework and use KeypoinyRCNN for keypoint detection, and we simply replace the backbone with RegionViT, and then output the local tokens at each stages as multi-scale features for the detector. The regional tokens are not used in the detection. Before sending the features to the detector framework, we add new layer normalization layers to normalize the features. We use RegionViT-S and RegionViT-B as the backbone and the weights are initialized from ImageNet1K pretraining. We train our models based on the settings in 1× schedule in Detectron2 (Wu et al., 2019), i.e., the batch size is 16 and the learning rate is set to 0.0001 which drops 10× at the 60,000-th and 80,000-th iteration. We use AdamW optimizer with weight-decay of 0.05, and resize the shorter and longer side of an image to 672 and 1,120, respectively. We also adopt drop-path when finetuning the detector, with a rate set to 0.2 for both RetinaNet and MaskRCNN. We trained the models with 8 GPUs. When training with longer schedule (3×), we adopt stronger data augmentation (multi-scale training) used in Swin (Liu et al., 2021) or Twins (Chu et al., 2021a) rather than fixed-size training used in 1× schedule.

## A.3 SEMANTIC SEGMENTATION

We adopt Semantic FPN as our semantic segmentation framework (Kirillov et al., 2019) by replacing the backbone network with RegionViT like the object detection task. We train the models with ImageNet1K pretrained weights. We mostly follow the training receipt in Twins (Chu et al., 2021a) The shorter side of the image is resized to 448 and the longer side won't exceed 1792, and then we crop a 448×448 region for training. During the evaluation, we take the whole image after resizing as illustrated above. We train the model with AdamW optimizer with learning rate 1e-4 for 80k iterations, and the learning rate is decayed with poly schedule (power is 0.9). The weight decay and drop-path rate is 0.05 and 0.2, respectively. We trained all models with 8 GPUs.

## A.4 ACTION RECOGNITION

We adopt the divided-space-time attention in TimeSformer (Bertasius et al., 2021) as the temporal modeling to perform action recognition experiments. More specifically, we plugin the temporal attention before every R2L transformer encoder and finetune the model pretrained with ImageNet1K (instead of ImageNet21K). We use the AdamW optimizer (Loshchilov & Hutter, 2019) with base linear rate of 2e-4 and weight-decay of 0.0001. We apply the same Mixup (Zhang et al., 2018), CutMix (Yun et al., 2019) and drop path (Tan & Le, 2019) in the image classification. For the input, we uniformly sample 8 frames from the whole video and the 224×224 region is cropped after shorter side of images is resized to the range of 256 to 320, resulting the size of the input video as 8×224×224. During evaluation, we use the same way to sample frames but take 3 224×224 spatial crops (top-left, center and bottom-right) after resizing shorter side of image to 256, and then ensemble the predictions as final prediction for the input video. We trained the models with 16 GPUs.

Table A3: Object detection performance on the COCO val2017 with 3× schedule. The bold number indicates the best number within the section, and for MaskRCNN, both $AP^b$ and $AP^m$ are annotated.

| Backbone | Params (M) | FLOPs (G) | RetineNet | | | | | | Params (M) | FLOPs (G) | MaskRCNN | | | | | |
| --- | --- | --- | --- | --- | --- | --- | --- | --- | --- | --- | --- | --- | --- | --- | --- | --- |
| | | | $AP$ | $AP_{50}$ | $AP_{75}$ | $AP_S$ | $AP_M$ | $AP_L$ | | | $AP^b$ | $AP^b_{50}$ | $AP^b_{75}$ | $AP^m$ | $AP^m_{50}$ | $AP^m_{75}$ |
| ResNet50 | 37.7 | 234 | 39.0 | 58.4 | 41.8 | 22.4 | 42.8 | 51.6 | 44.2 | 260.0 | 41.0 | 61.7 | 44.9 | 37.1 | 58.4 | 40.1 |
| PVT-S (Wang et al., 2021) | 34.2 | – | 42.2 | 62.7 | 45.0 | 26.2 | 45.2 | 57.2 | 44.1 | 245 | 43.0 | 65.3 | 46.9 | 39.9 | 62.5 | 42.8 |
| ViL-S (Zhang et al., 2021a) | 35.7 | 252.2 | 42.9 | 63.8 | 45.6 | 27.8 | 46.4 | 56.3 | 45.0 | 174.3 | 43.4 | 64.9 | 47.0 | 39.6 | 62.1 | 42.4 |
| Swin-T (Liu et al., 2021) | 38.5 | 245 | 43.9 | 64.8 | 47.1 | 28.4 | 47.2 | 57.8 | 47.8 | 264 | 46.0 | 68.2 | 50.2 | 41.6 | 65.1 | 44.8 |
| Twins-SVT-S (w/ PEG) (Chu et al., 2021a) | 34.3 | 209 | 45.6 | 67.1 | 48.6 | 29.8 | 49.3 | 60.0 | 44.0 | 228 | 46.8 | 69.2 | 51.2 | 42.6 | 66.3 | 45.8 |
| RegionViT-S | 40.8 | 192.6 | 45.8 | 67.2 | 49.2 | 30.0 | 50.0 | 60.5 | 50.1 | 171.3 | 46.3 | 68.8 | 50.6 | 42.3 | 65.5 | 45.7 |
| RegionViT-S+ | 41.5 | 204.2 | 46.9 | 68.3 | 50.7 | 31.1 | 51.0 | 62.0 | 50.9 | 182.9 | 47.3 | 69.5 | 52.0 | 43.1 | 66.4 | 46.7 |
| RegionViT-S+ w/ PEG | 41.6 | 204.3 | **46.7** | 68.2 | 50.2 | 30.7 | 50.8 | 62.4 | 50.9 | 183.0 | **47.6** | 70.0 | 52.0 | **43.4** | 67.1 | 47.0 |
| ResNet101 | 56.7 | 315 | 40.9 | 60.1 | 44.0 | 23.7 | 45.0 | 53.8 | 63.2 | 336 | 42.8 | 63.2 | 47.1 | 38.5 | 60.1 | 41.3 |
| ResNeXt101-32x4d | 56.4 | 319.1 | 41.4 | 61.0 | 44.3 | 23.9 | 45.5 | 53.7 | 62.8 | 340 | 44.0 | 64.4 | 48.0 | 39.2 | 61.4 | 41.9 |
| PVT-M (Wang et al., 2021) | 53.9 | – | 43.2 | 63.8 | 46.1 | 27.3 | 46.3 | 58.9 | 63.9 | 302 | 44.2 | 66.0 | 48.2 | 40.5 | 63.1 | 43.5 |
| ViL-M (Zhang et al., 2021a) | 50.8 | 338.9 | 43.7 | 64.6 | 46.4 | 27.9 | 47.1 | 56.9 | 60.1 | 261.1 | 44.6 | 66.3 | 48.5 | 40.7 | 63.8 | 43.7 |
| Swin-S (Liu et al., 2021) | 59.8 | 335 | 46.3 | 67.4 | 49.8 | 31.1 | 50.3 | 60.9 | 69.1 | 354 | 47.6 | 69.4 | 52.5 | 42.8 | 66.5 | 46.4 |
| Twins-SVT-B (w/ PEG) (Chu et al., 2021a) | 67.0 | 322 | 46.9 | 68.0 | 50.2 | 31.7 | 50.3 | 61.8 | 76.3 | 340 | 48.0 | 69.5 | 52.7 | 43.0 | 66.8 | 46.6 |
| RegionViT-B | 83.4 | 308.9 | 46.1 | 67.8 | 49.1 | 31.5 | 50.2 | 61.2 | 92.2 | 287.9 | 47.2 | 69.1 | 51.7 | 43.0 | 66.4 | 46.5 |
| RegionViT-B+ | 84.4 | 328.1 | 46.9 | 68.6 | 50.1 | 30.8 | 50.7 | 62.6 | 93.2 | 307.1 | 48.1 | 70.2 | 52.5 | 43.5 | 67.1 | 47.1 |
| RegionViT-B+ w/ PEG | 84.5 | 328.2 | **46.9** | 68.3 | 50.3 | 31.1 | 50.5 | 62.4 | 93.2 | 307.2 | **48.3** | 70.1 | 52.8 | **43.5** | 67.1 | 47.0 |
| ViL-B (Zhang et al., 2021a) | 66.7 | 443.0 | 44.7 | 65.5 | 47.6 | 29.9 | 48.0 | 58.1 | 76.1 | 365.1 | 45.7 | 67.2 | 49.9 | 41.3 | 64.4 | 44.5 |
| RegionViT-B+ w/ PEG† | 84.5 | 506.4 | **48.2** | 69.9 | 51.5 | 34.3 | 51.5 | 61.7 | 93.2 | 464.4 | **49.2** | 71.0 | 53.7 | **44.5** | 68.4 | 48.3 |

The reported results of Swin (Liu et al., 2021) are from Twins (Chu et al., 2021a) as the original paper does not include resutls with ImageNet1K weights. †: input resolution is 896×1344.

Table A4: Performance on person keypoint detection.

| Model | FLOPs (G) | Params (M) | bbox AP (%) | keypoint AP (%) |
| --- | --- | --- | --- | --- |
| ResNet-50 | 137.7 | 59.1 | 53.6 | 64.0 |
| RegionViT-S+ (w/ PEG) | 172.2 | 65.7 | **56.0** | **66.1** |

# B  MORE DETAILED RESULTS

We provided more details of some Tables shown in the main paper here. Table A2 and Table A3 include more metrics as compared to Table 5. Table A4 shows the results on keypoint detection. RegionViT outperforms ResNet-50 with moderate increases in FLOPs and parameters. This suggests that RegionViT can model the global context in keypoint detection as well. Table A5 includes complexity comparison for the ablation study on regional tokens as shown in Table 8.

Table A5: Performance w/ and w/o regional tokens on RegionViT-S.

| Regional Tokens | FLOPs (G) | Params (M) | ImageNet1K Acc. (%) |
|---|---|---|---|
| N | 5.0 | 30.4 | 82.2 |
| Y | 5.3 | 30.6 | 82.6 |
| MaskRCNN | | | MS COCO ($AP^b$/$AP^m$) |
| N | 168.5 | 49.9 | 40.5/37.8 |
| Y | 171.3 | 50.1 | 42.2/39.0 |
| RetinaNet | | | MS COCO ($AP^b$) |
| N | 189.8 | 40.5 | 40.7 |
| Y | 192.6 | 40.8 | 41.7 |
| SemanticFPN | | | ADE20K (mIoU) |
| N | 36.8 | 34.7 | 42.3 |
| Y | 37.3 | 35.0 | 43.7 |

Table A6: Throughput comparison.

| Model | Acc. (%) | Params (M) | FLOPs (G) | Throughput (images/sec) |
|---|---|---|---|---|
| Swin-T | 81.3 | 29.0 | 4.5 | 1129 |
| Swin-S | 83.0 | 50.0 | 8.7 | 710 |
| RegionViT-S | 82.6 | 30.6 | 5.3 | 823 |
| RegionViT-M | 83.1 | 41.2 | 7.4 | 706 |

