# OpenReview forum: "RegionViT: Regional-to-Local Attention for Vision Transformers"
_ICLR.cc/2022/Conference — ICLR 2022 Poster_

### Official Review · Reviewer_rREV · 2021-11-02

**Correctness:** 3
**Technical Novelty And Significance:** 3
**Empirical Novelty And Significance:** 3
**Recommendation:** 6
**Confidence:** 3

**Main Review:**

+ The performance of this work is promising.
-  The novelty of this work is somewhat trivial, the issue of the interaction between global and local information has been proposed by previous work, not only in image classification but also in other downstream tasks. And the proposed methods are not novel enough when compare with the previous ones.
- Several grammatical mistakes are made. e.g. In abstract, the regional self-attention extract global information among... (extracts);


**Summary Of The Paper:**

This paper proposes a new architecture that adopts the pyramid structure and employs a novel regional-to-local attention for vision applications. The regional-to-local attention reduces the memory complexity as compared to standard self-attention.


**Summary Of The Review:**

This work aims to solve the interaction between global and local information in a ViT architecture and to save the computation at the same time, the performance is good. However, I am concerned with the novelty of this work and the contribution may be limited.

---

> ### Author Response · Authors · 2021-11-23
> **Response to Reviewer rREV**
>
> Thanks for the constructive comments. We have polished the paper and fixed many grammatical mistakes to make it more readable.
>
> Regarding technical novelty, we propose a new vision transformer (RegionViT) based on regional-to-local attention to learn both local and global features. Our proposed regional-to-local attention alleviates the overhead of standard global attention (too many tokens) and the weakness of pure local attention (no interaction between regions) used in existing vision transformers. Our models outperform or are on par with several **concurrent** works on vision transformer that exploit the pyramid structure for image classification. We differentiate our work from many recent works in detail in Table 1, e.g., Swin, ViL, and Twins. Below we provide a more detailed comparsion between our approach and ViL/Twins which are also based on global + local concepts.
>
> **Difference from Twins (NeurIPS 2021).** The two major differences lie at *(a) How to perform the global attention.* Twins uses global sub-sampled attention to perform global attention. In contrast, we use the regional tokens to model global information and then pass it to local tokens via Local Self-Attention (LSA). *(b) Local self-attention.* In Twins, the local self-attention only performs on the local tokens within a window. However, our LSA includes one extra spatially-corresponded regional token which enables the LSA to simultaneously get both local and global information efficiently thanks to the Regional Self-Attention (RSA). With those differences, our approach achieves better performance with fewer model parameters.
>
> **Difference from ViL (ICCV 2021).** The two major differences are: (a) the global token in ViL is location-independent while our regional tokens are location-aware and ViL only uses one global token which can not model the hierarchy of an image. (b) ViL uses an overlapped sliding window to perform local attention while ours is based on a non-overlapped approach. Thus, our approach results in better performance than ViL.

---

### Official Review · Reviewer_1u9G · 2021-11-02

**Correctness:** 3
**Technical Novelty And Significance:** 3
**Empirical Novelty And Significance:** 3
**Recommendation:** 6
**Confidence:** 4

**Main Review:**

**Strengths:**
- Overall the paper is fairly easy to follow and it's easy to understand how the approach works, including a fair amount of detail to potentially reproduce results. The provided code, albeit I did not run it, seems like it should work.
- In many of the presented experiments, RegionViT does indeed obtain on par or state-of-the-art results when compared to similarly sized models
- Quite some relevant and interesting ablations are presented.

**Weaknesses:**
- For me the main weakness of the paper is that it seems fairly incremental in a longer line of ideas. First there was ViT, followed by more efficient training from DeiT including many augmentations, several works then proposed a pyramid-like architectures, and finally this work is a small increment on how to best realize the pyramid architecture including some smaller tweaks. The scores indeed do improve, but sometimes there are quite some different variants (vanilla, +, + w/PEG) that are not consistently listed for the different tasks, partially raising the question how large the margin to the previous models really is. While I don't doubt that the approach has some merit, I would find the paper significantly more valuable if the focus of the experiments would be to dive deeper into how to best design the pyramid structure of the network and less on listing many different tasks with some table showing state of the art results. Now this paper presents yet another new proposal on how to change change the ViT architecture, but I gain little insight into why it really performs better. Shifting the focus to this part of the paper would be more interesting in my opinion. For example it would be interesting to discuss some of the ablation results in more details, or to do a more thorough investigation into the size of the local and global patches. One could even ask the question if two levels (local+regional) are optimal, or an additional super-regional or global attention could also be useful?
- I was a bit surprised by the fairly low results from the ImageNet 21k transfer experiments. Of course it is clear that the FLOP count and parameter count is significantly lower than the better models in Table 4a, however, it does raise the question if RegionViT can indeed be scaled to such an extend that it matches the state of the art. Given that one big part of method is the reduced complexity, it is also not surprising that the model is computationally less heavy, but it would be a drawback if there is some performance ceiling it hits due to the specific architecture. Here it would be good to create some even bigger model to see if the larger models can be matched or outperformed. Also the other transfer experiments in Table 4 are not super convincing.
- The keypoint detection and action recognition sections take up quite some space and especially Table 6 does not list any convincing baselines. At the same time these experiments don't add huge value to the overall story in my opinion. I would recommend to use this space to perform more relevant and interesting ablations/experiments/discussions instead and move these to the supplementary material.
- Lastly, the writing is not the best in some parts of the paper. For a potential final version, I would recommend that the paper is proofread by a native speaker.


**Open questions:**
- Table 1 makes some statements about overlapping vs non-overlapping patches. The way the local tokenization works (using 1 or 3 convs)), I would say it is somewhat of a stretch so say that the patches are strictly non-overlapping.
- I was fairly surprised about weights being shared between RSA and LSA, as well as in the downsampling convs. I don't really see a larger number of parameters as a huge issue for a model and the FLOP count/inference speed should be the same if these weights would not be shared. Actually ablating this would be very interesting. I would kind of expect that the local and regional tokens contain slightly different things and having separate weights could potentially improve the results?
- The paper states that this model saves 73% of memory. I'm wondering if this is an empirical measure, or it was derived based on the complexity analysis? I guess this factor might also be different for the Ti/S/B/L models?
- I find it quite surprising that the regional tokens don't have a huge effect on the overall performance (Ablation 1). Is the remaining strong performance based on the pyramidal structure of RegionViT? Given that the pure local attention never is performed on overlapping areas that is the only reason I would see why the model still remotely works.

**Summary Of The Paper:**

The paper introduces a new spin on the ViT architecture. Similar to other recent developments, it proposes a pyramidal style architecture where subsequent layers process the input image at decreasing resolutions. This allows the approach to be significantly more memory and compute efficient compared to the original ViT architecture, while still obtaining solid results on several tasks and benchmarks. At the core of the method is a regional-to-local attention approach, where very coarse regional tokens are processed by a normal multi-head attention yielding global attention and each coarse regional patch is additionally represented by a set of smaller local patches, which are also processed together with the corresponding regional token to get finer-grained local attention. Each layer in the network follows this principle and between different sections of the network both the regional and local tokens are downsampled with strided conv layers.

**Summary Of The Review:**

Overall the paper proposes a fairly interesting pyramidal modification to ViT. It seems like a somewhat incremental approach though. While the performance is empirically shown, there is little that can really be learned about the underlying approach from the experiments. Nothing is really wrong with the paper, I do think it could be significantly better though. As such I recommend the paper to be accepted, but I do think it has a lot more potential that is not uncovered at the moment and I would actually be very happy to read such an improved version of the paper.

---

> ### Author Response · Authors · 2021-11-23
> **Response to Reviewer 1u9G**
>
> We thank the reviewer for acknowledging that our RegionViT obtains on par or state-of-the-art results when compared to similarly-sized models. We have incorporated all your suggestions including new experimental results in the revised version.
>
>
> **(a) Deep dive with more ablation studies.** We have added new ablation studies suggested by the reviewer as well as the one suggested by reviewer YwKN (using all regional tokens as keys in LSA) to our paper. We believe that these new ablation studies will help the reader better understand the design choices of our work.
>
> **(b) Different variants and more hierarchies.** In order to compare with other works more fairly, especially for the downstream tasks, we select the models that result in similar complexity. On the other hand, since we found that PEG is helpful for the dense prediction tasks, we do show the models with and without PEG in object detection to justify the contributions of PEG.
>
> We agree with the reviewer that exploring more hierarchies is indeed an interesting future direction. We will explore this direction, including the number of hierarchies, the patch size for each level and the window size at each level.
>
> **\(c\) Performance on ImageNet21K.** Due to limited computing resources, we are not able to train a larger RegionViT (e.g., deeper or wider) on ImageNet21k; however, when we compare to the model under similar complexity, e.g. ViL-B (shown in both ImageNet1K and ImageNet21K), our model performs consistently better in terms of performance. This suggests that our models should be able to scale to larger datasets.
>
> We follow the prior works [1, 2] and perform transfer learning experiments to validate the transferability of our approach to other image classification tasks.
>
> **(d) Results on keypoint detection and action recognition.** Thanks for the suggestion. We have moved the keypoint detection results (Table 6) to the appendix and simplified the action recognition section in the revised paper.
>
> **(e) Weight Sharing.** Based on the reviewer's suggestion, we conduct an experiment without parameter sharing between RSA and LSA as well as downsampling in our approach. RTable 1 shows the results (added as Table 10 of the revised paper).
>
> RTable 1. Ablation study w/ and w/o parameter sharing between RSA and LSA as well as downsampling on ImageNet1K. Model is RegionViT-S.
>
> | Weight sharing | FLOPs | Parameters | Top-1 Acc. |
> | -------- | -------- | -------- | ---|
> | N | 5.3 | 40.4 | 82.7 |
> | Y | 5.3 | 30.6 | 82.6 |
>
> As seen from the table, using separated weights for RSA and LSA only slightly improves the accuracy but significantly increases the model size. This suggests sharing parameters between RSA and LSA suffices for learning local and global information in our approach.
>
> **(f) Regional tokens and overlapping windows.** Even though the regional tokens only contribute moderate improvement for the classification task, it is shown to be helpful for downstream tasks such as object detection and semantic segmentation. This is because dense prediction tasks require more features with global contextual information (provided by the regional tokens) than image classification.
>
> **(g) Memory savings.** The 73% memory savings are estimated based on the main component of our approach (stage 3 in Figure 2 of the paper) when comparing to standard multi-head self-attention with 196 tokens (the same number of tokens of the standard ViT with patch size 16). Note that each stage and each model will have different memory savings.
>
> **(h) Clarification on a non-overlapped statement in Table 1.** Thanks for the suggestion! We have updated Table 1 in the revised paper by changing Swin to a strictly non-overlapped method while Twins is not strictly non-overlapped since it uses the PEG to mix tokens.
>
> **References**:
> [1] Cross-Attention Multi-Scale Vision Transformer for Image Classification, Chun-Fu Chen, Quanfu Fan, Rameswar Panda, ICCV 2021.
> [2] Training data-efficient image transformers & distillation through attention, Hugo Touvron, Matthieu Cord, Matthijs Douze, Francisco Massa, Alexandre Sablayrolles, Hervé Jégou, ICML 2021.

---

### Official Review · Reviewer_iQks · 2021-11-02

**Correctness:** 3
**Technical Novelty And Significance:** 2
**Empirical Novelty And Significance:** 2
**Recommendation:** 6
**Confidence:** 3

**Main Review:**

### Advantages
1) This paper is well written and easy to follow.
2) The experiments are sufficient to evaluate the proposed method.

### Disadvantage
Since the R2L attention has two steps, I am concerned about the throughput of RegionViT. FLOPs can not directly reflect the throughput. It is better to show the throughput comparison for Table 3.

**Summary Of The Paper:**

This paper introduces local inductive bias into vanilla vision transformer by adopting the pyramid structure and employing regional-to-local attention. The regional-to-local (R2L) attention is the main contribution of the paper. Compared to the vanilla self-attention, R2L has two steps: 1) self-attention on regional tokens; 2) self-attention on local tokens and their corresponding regional token.

**Summary Of The Review:**

The idea sounds direct and reasonable, although introducing local inductive bias into ViT is not and there have been lots of paper to achieve it. The paper is well-written and shows efficient experiments to evaluate the proposed RegionViT.

---

> ### Author Response · Authors · 2021-11-23
> **Response to Reviewer iQks**
>
> Thanks for recognizing our work! Below, we compare the throughput of our work with Swin transformer. We use the authors' released codes of Swin to compute the inference throughput on an Nvidia V100 GPU (32GB) with batch size 128.
>
> | Model | Acc. (%) | Params (M) | FLOPs (G) | Throughput (images/sec) |
> | --- | --- | --- | --- | --- |
> |Swin-T | 81.3| 29.0 | 4.5  | 1129 |
> |Swin-S | 83.0| 50.0 | 8.7  | 710 |
> |RegionViT-S  | 82.6 | 30.6 | 5.3  | 823 |
> |RegionViT-M  | 83.1 | 41.2 | 7.4  | 706 |
>
> Our throughput is comparable to Swin, while RegionViT-M is more parameter efficient with similar performance. Note that the throughput is highly dependent on computing platforms and whether or not the operations are optimized. E.g., PyTorch recently supports `CUDA Graphs` to accelerate computation, and this feature can potentially speed up our method further as we can encapsulate entire R2LAttention into one CUDA kernel reducing the overhead of two steps. https://pytorch.org/blog/accelerating-pytorch-with-cuda-graphs/

---

### Official Review · Reviewer_YwKN · 2021-11-02

**Correctness:** 3
**Technical Novelty And Significance:** 3
**Empirical Novelty And Significance:** 2
**Recommendation:** 6
**Confidence:** 4

**Main Review:**

+ives
The paper is well-written and easy to follow.
The experimental evaluation on various dataset is very impressive. However, the accuracy improvement is not convincing.
The accompanied source code.

-ives
There is a significant advancement using part-hierarchies (Zheng et al TIP 2019), Attention Pyramid (Ding et al. TIP 2021), attention-driven hierarchical multi-scale (Warton et al., ArXiv 2021, I am aware that this paper is added recently)  and patches are all you need (this is also under review, https://github.com/tmp-iclr/convmixer). It is unclear how the regions and patches are novel. To me, it is how adapted in transformers.

It is unclear how memory saving is 73% when you are computing self-attention among regions followed by local self-attention between regions and local patches.

Is dimension of C for features representing local patches and regional patches same?

The ablation studies is very comprehensive. However, missing ablation study without self-attention between regions (first step of R2L attention)

It is also unclear on the 2nd step of the which one is used as key for multi-head attention is it  local patches? If so what could be the outcome if we use the region as keys.

The overall accuracy of the model is good but not significantly better than the SotA approaches. The accuracy is better for + models in which the window size is 14 and the model has higher GFLOPS and parameters.

**Summary Of The Paper:**

The paper presents a new ViT architecture by adopting the concepts of pyramid structure and employ regional-to-local attention instead of global self-attention as in standard vision transformers. The regional and local concept is linked to patch size i.e. region consists of larger patches in comparison to local. A regional token is associated a set of non-overlapping local tokens while computing reginal-to-local attention. This attention is computed using a regional self-attention comprising all reginal tokens to extract global information, followed by a local self- attention mechanism that pass information between the reginal token and the linked local tokens via self attention. The approach is evaluated image recognition, object and keypoints detection, semantic segmentation and action recognition. The performance is comparable to the state-of-the-art.

**Summary Of The Review:**

The idea of large patches to capture high-level shape, as well as concentrate on detailed texture and parts information using smaller patches is very good. However, the justification could have been improved. The paper is well-written and the experimental evaluation is comprehensive. However, the improvement in accuracy is not significant in comparison to the way the novelty of R2L attention is described. There is an advancement to the transformer model however the impact is limited. The related work is very nicely done.

---

> ### Author Response · Authors · 2021-11-23
> **Response to Reviewer YwKN (1/2)**
>
> We thank the reviewer for the insightful questions and great suggestions. We have revised the paper by incorporating all the changes including citations to the new related references.
>
> **(a) Difference from existing works on regions and patches**. Below are the key differences between our work and the reviewer's referred works:
>
> **Difference from Zheng et al. TIP 2019 [1].** Zheng et al. propose a progressive approach to refine regional of interest by attention for fine-grained image classification; however, our RegionViT does not refine the regions progressively, it directly models multi-scale features by R2L attention. Moreover, RegionViT is orthogonal to Zheng's approach, we can further apply their approach on top of RegionViT for the fine-grained recognition.
>
> **Difference from Ding et al. TIP 2021 [2].** This paper proposes a pyramid structure with top-down and bottom-up pathways to enhance/select features and compose regions of interest for fine-grained visual recognition. In feature fusion, they simply align the feature maps at different levels and sum them up. Our R2LAttention on the other hand, sends a single regional token to LSA and hence the local tokens can get global information more efficiently compared to the approach in Ding et al. [2].
>
> **Difference from Wharton et al. BMVC 2021 [3].** This paper models multi-level features by using a computationally intensive graph convolutional network for the fine-grained recognition; while we propose R2LAttention to efficiently fuse the multi-scale features from the regional and local tokens. Note that our approach can also be combined with [3] to extend RegionViT for fine-grained recognition.
>
> **Difference from Patches Are All You Need [4].** This is a very recent paper which proposes an alternative of multi-head self-attention. Our work is orthogonal to [4] as the key idea of RegionViT is on efficiently communicating the information between global and local regions. We can replace the multi-head self-attention in RSA and LSA with ConvMixer [4].
>
> To summarize, our RegionViT proposes a simple yet effective approach for communicating the global and local information by using R2LAttention that enables the local tokens to get global information in an efficient way. Extensive experiments on dense-prediction tasks, clearly demonstrate that the regional tokens can improve the performance without the significant overhead (~2% for detection, 1.5% for segmentation, Table 8 of the revised paper).
>
>
> **(b) Clarification on memory savings.** The saving comes from the local self-attention when compared to the standard self-attention. The memory complexity of a standard multi-head self-attention is O(N^2), where N is the number of tokens. Now, local self-attention confines the attention in a local region with M tokens, therefore, the complexity of each local region becomes O(M^2) and there are (N/M) regions. Then, the overall complexity is about O(NM). Thus, when using window size 7, at the main stage of our RegionViT, (stage 3), where M is 49 and N is 196, the memory complexity saving is 75% but with the regional tokens, the savings go down slightly to 73%. We have made this clear in the revised paper.
>
>
> **\(c\) Ablation study without self-attention between regions (first step of R2L attention).** We have a similar ablation study on this to verify the role of regional tokens (Table 8 of the revised paper, w/ and w/o regional token). Table 8 of the revised paper discusses the importance of the regional tokens, i.e., either to keep or remove the regional tokens. Thus, when the regional tokens are removed, the RSA is also removed. The results show that the regional tokens plays the important role in dense-prediction tasks.
>
> Following the reviewer's suggestion, we also conduct an additional experiment by keeping the regional tokens but skipping RSA. This model achieved similar results to the ablation study in Table 8 of the revised paper, once again showing the importance of RSA in communicating global and local information.
>
> **(d) Is the dimension of C for features representing local patches and regional patches the same?** Yes, the dimension of the two tokens is the same as shown in Figure 2.
>
> **References:**
> [1] Learning Rich Part Hierarchies With Progressive Attention Networks for Fine-Grained Image Recognition, Heliang Zheng, Jianlong Fu, Zheng-Jun Zha, Jiebo Luo, and Tao Mei, IEEE TIP 2020.
> [2] AP-CNN: Weakly Supervised Attention Pyramid Convolutional Neural Network for Fine-Grained Visual Classification, Yifeng Ding,
> Zhanyu Ma, Shaoguo Wen, Jiyang Xie, Dongliang Chang, Zhongwei Si, Ming Wu and Haibin Ling, IEEE TIP 2021.
> [3] An attention-driven hierarchical multi-scale representation for visual recognition, Zachary Wharton, Ardhendu Behera, Asish Bera, BMVC 2021.
> [4] Patches Are All You Need?, ICLR 2022 submission.

---

> > ### Author Response · Authors · 2021-11-23
> > **Response to Reviewer YwKN (2/2)**
> >
> > **(e) It is also unclear on the 2nd step of the which one is used as key for multi-head attention is it local patches? If so what could be the outcome if we use the region as keys.** The keys and values used in LSA are 1) one regional token and 2) the corresponding local tokens as shown in Eq. 2 top-right ($y^d_{i,j}$). Regarding the ablation (i.e., use of regional tokens as keys) suggested by the reviewer, if we only use the regional tokens as the keys in LSA, that means there is no interaction between the local tokens, which seems not to be intuitive. Therefore, we conduct another experiment that includes all the regional tokens as well as the local tokens as the keys (instead of a single token that is associated with the current local tokens). This model achieves 0.1% improvement over RegionViT-S but it results in 13% more computations (0.7 GFLOPs), which suggests that when considering the regional tokens, the one associated with the current region is sufficient to achieve good performance (See Table 11 of the revised paper).
> >
> >
> > **(f) The overall accuracy of the model is good but not significantly better than the SotA approaches. The accuracy is better for + models in which the window size is 14 and the model has higher GFLOPS and parameters.** Our work is marginally better or on par with those **concurrent** SotA approaches, while requiring either fewer FLOPs or parameters, e.g. when comparing to Twins (accepted in NeurIPS 2021), our model parameters are fewer. Even with window size 14, our models do not increase the FLOPs significantly, e.g., RegionViT-B+ achieves comparable accuracy with fewer FLOPs and parameters than Twins-SVT-L.

---

### Author Response · Authors · 2021-11-23
**Summary of Author's Response and Paper Revision**

We thank all the reviewers for their constructive comments! We are encouraged that reviewers find that our paper is well-written and easy to follow (YwKN, iQks and 1u9G), our experimental evaluation on various dataset is very impressive (YwKN) and performance is promising (rREV, 1u9G) and the results are sufficient to evaluate the proposed method (iQks). Moreover, the detailed description and codes facilitate the reproducibility (1u9G).

We have addressed all the concerns that the reviewers posed with additional experimental comparisons and clarifications. All of these additional experiments and suggestions have been added to the updated PDF (changes are highlighted in red). Below, we summarize the summary of the response and encourage the reviewers to take a look at the new additions.
- Additional results on using all regional tokens in LSA, as suggested by R-YwKN (Table 11 of the revised paper).
- Additional results on not sharing the weights between RSA and LSA as well as downsampling, as suggested by R-1u9G (Table 10 of the revised paper).
- Clarification on the memory saving section, as suggested by R-YwKN and R-1u9G.
- Reorganization of result section: merging object detection and semantic segmentation, moving keypoint detection to the appendix, and simplifying the section of action recognition, etc.

---

> ### Comment · Reviewer_1u9G · 2021-11-24
> **Thanks for the rebuttal**
>
> Thank you for the fairly lengthy discussion of all the comments.
>
> Based on this, the paper has not been changed significantly though. I do think it has become a little bit better, but my overall opinion of the paper has not changed a lot. Looking at all the other reviews and the rebuttal, I would say that at least from my side I'm okay with the paper being published, but I guess it might remain a close call depending on all other submissions and what acceptance rate is targeted for ICLR 2022.

---

> ### Comment · Reviewer_YwKN · 2021-11-29
> **Many thanks for the rebuttal**
>
> Thank you for addressing many raised concerns.
>
> However, the answers to the questions (e) and (f) are not convincing. I have gone through the other reviewers comments and the authors response to those. The updated version of the paper is bit better but "not significantly" impacted on my decision about this paper. Thus, I am sticking to my original decision on "6 - marginally above the acceptance threshold".

---

### Decision · Program_Chairs · 2022-01-20

**Decision:**

Accept (Poster)

**Comment:**

The paper proposed a new architecture called Regional-to-Local attention for the vision transformers. The idea is easy to understand, the model adopts the pyramid structure and adds a regional to local attention instead of using the global attention. The architecture is well-motivated and the paper is generally well written.

The main concerns from the reviewers are mostly clarification questions. The authors did a good job addressing them. Apart from those, most reviewers raise the novelty issue of such architecture, which I would think is a drawback of this paper.

I am leaning towards the acceptance of this paper mainly because of its experimental results. It is the best in my batch and I think there is a significant improvement over the previous approaches.